# High-Precision Real-Time Detection of Blast Furnace Stockline Based on High-Dimensional Spatial Characteristics

**DOI:** 10.3390/s22166245

**Published:** 2022-08-19

**Authors:** Pan Liu, Zhipeng Chen, Weihua Gui, Chunhua Yang

**Affiliations:** School of Automation, Central South University, Changsha 410083, China

**Keywords:** blast furnace, stockline detection, mechanical probe, radar probe, high-dimensional spatial features

## Abstract

The real-time, continuity, and accuracy of blast furnace stockline information are of great significance in reducing energy consumption and improving smelting efficiency. However, the traditional mechanical measurement method has the problem of measuring point discontinuity, while the radar measurement method exhibits problems such as weak anti-interference ability, low accuracy, and poor stability. Therefore, a high-dimensional, spatial feature stockline detection method based on the maximum likelihood radial basis function model (MLRBFM) and structural dynamic self-optimization RBF neural network (SDSO-RBFNN) is proposed. Firstly, the discrete time series joint partition method is used to extract the time dimension periodic features of the blast furnace stockline. Based on MLRBFM, the high-dimensional spatial features of the stockline are then obtained. Finally, an SDSO-RBFNN is constructed based on an eigen orthogonal matrix and a right triangular matrix decomposition (QR) direct clustering algorithm with spatial–temporal features as input, so as to obtain continuous, high-precision stockline information. Both the simulation results and industrial validation indicate that the proposed method can provide real-time and accurate stockline information, and has great practical value for industrial production.

## 1. Introduction

In the iron and steel industry, energy saving and emission reduction are the main challenges. In steel production, the ironmaking process consumes more than 70% of the energy and emits most of the CO_2_ [1]. The blast furnace stockline is a key control parameter in the ironmaking process. Accurate and continuous real-time blast furnace stockline information is the key to adjusting the blast furnace production process, achieving precise control of blast furnace material charging, and stabilizing blast furnace working conditions [2]. A blast furnace is a typical black box and a large metallurgical reactor [3]. The environment in the furnace is extremely harsh, with the features of high temperature, high pressure, closed, and dusty, etc. [4]. It is very difficult to accurately measure the blast furnace stockline in real time without interruption [5]. At present, mechanical probes and radar probes are common tools for stockline detection [6]. The mechanical probe method is to use a steel wire rope to lower the heavy hammer to the blast furnace burden surface, and measure the length of the steel wire rope through the winch to obtain the stockline information. The contact-type stockline measurement method has high accuracy and stability [7], but there are defects, such as a lengthy measurement cycle, the inability to continuously measure, and being unable to perform any work during the blast furnace material feeding period [8]. As another important piece of measuring equipment for the blast furnace stockline, the radar probe receives the directional radar wave reflected from the material surface, processes it with the time-of-flight (TOF) algorithm, and measures the stockline information in real-time. This method is widely used due to its advantages of non-contact, high penetration, and real-time performance [9]. However, because the smelting environment in the blast furnace is extremely harsh [10], the high-speed airflow and high concentration of dust in the furnace often interfere with the radar directional echo signal, which makes the measurement accuracy fluctuate, and have poor stability and low reliability [11]. Therefore, overcoming the defects of the two methods for stockline measurement and using the advantages of the two methods, so as to obtain the stockline detection data continuously and with high precision in real-time, is a scientific problem that requires solving urgently.

In order to improve the stability of the traditional mechanical probe and enhance its adaptability to abnormal working conditions, Ma et al. proposed a closed-loop speed control method, which realized anti-flip control by changing the anti-flip torque in the variable torque control, so as to realize the uniform motion of the mechanical probe weight hammer, and ensure the stability and accuracy of the process [12]. Liu proposed a new encoder–decoder architecture composed of a convolutional neural network (CNN) and a long-term and short-term memory (LSTM) network. This architecture suppressed the noise interference in the measurement of the mechanical probe. Based on an LSTM network, a large number of historical measurement data were modeled, and the model was used to predict the unmeasured material sites of a mechanical probe to solve the problem of discontinuous measurement [13]. Although the above research improves the reliability and accuracy of measuring the stockline with a mechanical probe, it cannot realize high-precision and continuous real-time measurement with a mechanical probe. In the research of measuring a blast furnace stockline with a radar probe, Wang based on the learning-based key points estimation (KP-BSP) method, reconstructed the key points in the BSP image of a radar probe, and proposed the key-points-based connected region noise reduction (KP-CRNR) algorithm to eliminate the influence of noise, improve the signal-to-noise ratio of the radar signal, and the measurement accuracy of the radar probe [14]. An improved solid-state radar measurement and signal processing method were proposed in [15], and a special phase-controlled radar was designed, which adopted the improved FM continuous wave measurement principle, and combined the intelligent time-varying threshold signal processing method to synchronously improve the real-time performance and accuracy of stockline measurement. Yu improved the three-dimensional synthetic aperture radar (SAR) imaging method of a blast furnace stockline based on a range migration algorithm (RMA), and increased the imaging speed and imaging quality by adding raw data down-sampling, targeting feature extraction, frequency domain zero filling, and other operations [16]. Xiu adopted a threshold segmentation method based on bandwidth variance iteration to remove noise, and an energy center of gravity method based on weighted sampling to sharpen the peak ridge, which improved the accuracy of the swing radar measurement of blast furnace stockline data [17]. The above methods were breakthroughs in radar probe principle design and data processing algorithms, which provide technical support for radar imaging of the whole blast furnace charge surface, and are of great significance for fine control of blast furnaces [18]. However, the problems of low measurement accuracy, large fluctuation, and the instability of the radar probe ruler have not been solved [19]. At present, more researchers are focused on building an intelligent model based on the detection data of the stockline collected by two probes through a deep network, which integrates the advantages of two measuring instruments, and realizes high-precision, continuous soft-sensing of the stockline. For example, in [20], Chen et al. proposed the dynamic self-growing RBFNN, based on the data distribution features of a radar probe, which combines the advantages of two probes online in real-time, improves the accuracy of stockline measurement on the premise of ensuring the continuity of stockline measurement, and has certain practical value. In [21], Chen et al. extracted the spatiotemporal features of radar probe data, and based on the fast clustering algorithm of feature vector space, constructed the efficient structure self-tuning RBFNN, and realized the fusion measurement of two kinds of probe.

The above-mentioned detection methods realize the fusion measurement of radar probe data and mechanical probe data. However, it is difficult to extract the spatial distribution parameters of a radar probe, and the clustering results are greatly affected by the initial value, which still needs to be solved.

For this reason, a real-time online measurement method for the blast furnace stockline, based on high-dimensional spatial features, is proposed. Figure 1 shows the research framework. Firstly, the discrete time series joint partition method is used to divide the joint period of radar and mechanical probe data. Then, aiming at the high-order, non-linear features of radar probe data space in a period, a maximum likelihood radial basis function model modeling method is proposed, which fully excavates the internal structure of the radar probe data and extracts the high-dimensional spatial features. On this basis, taking the high-dimensional spatial features and periodic features of the radar probe as inputs, based on the proposed QR decomposition direct clustering algorithm, a structural dynamic self-optimization RBF neural network is constructed. By integrating the advantages of mechanical probes and radar probes, the real-time continuous and high-precision detection of stockline information is obtained.

The remainder of the paper is organized as follows: In Section 2, the high-dimensional spatial features extraction of radar data, based on the maximum likelihood radial basis function model, is described. In Section 3, a structural dynamic self-optimizing RBF neural network, based on feature QR decomposition and a direct clustering algorithm, is proposed for the real-time, high-precision detection of a blast furnace stockline. In Section 4, the simulation and industrial verification results are given. Section 5 is the conclusion.

## 2. High-Dimensional Spatial Feature Extraction

### 2.1. Data Analysis and Material Charging Cycle Division

In the process of blast furnace material charging, the iron ore and coke are unloaded to the blast furnace burden surface in batches through the rotary chute. According to the material charging technology, the material charging process is usually divided into the material feeding period and the material charging idle period (abbreviated as idle period), in which the material feeding period lasts about 1.5–2 min and the idle period lasts about 1.5–4 min. Under normal working conditions, the duration of a complete material charging cycle of blast furnace is usually controlled within 3–6 min. Figure 2 shows the stockline information (the height of stockline) obtained by radar and mechanical probes, which shows that the change in stockline exhibits periodic features. The stockline decreases with the increase in burden surface in the material feeding period, while the stockline increases with the decrease in burden surface in the idle period. As shown in Figure 2, it is also found that the sampling process of the stockline by the mechanical probe is discrete. It is only sampled once in the idle period of each material charging cycle, while the sampling of the stockline by the radar probe is once every 10 s, which leads to the sampling data of the two probes being asynchronous, and it is difficult to match them. Therefore, a sliding window data processing method is presented, as shown in Figure 3. A window with width N is used to map radar data with the mechanical probe data. N can be determined according to the degree of correlation between the two detected data. Through the sliding operation, the matching problem of the infinite dimensional data of the two detection methods is transformed into the corresponding problem of N data in one window, which lays a foundation for the deep mining of the internal relationship between the two detection data and real-time accurate measurement of stockline.

Considering that the material charging operation of a blast furnace is carried out periodically, the high-precision periodic division of the radar and mechanical probe data is the key to mining the internal relationship between the data of the two probes and extracting the high-dimensional spatial features of blast furnace stockline change. Setting the radar probe data as a finite set of time series is described as:(1)$$R={\{\left(t_{1},l_{1}\right),\left(t_{2},l_{2}\right),\cdots ,\left(t_{p},l_{p}\right)\}|}_{t_{j}<t_{j+1}\left(j=1,2,\cdots ,p-1\right)}$$
where p is the number of radar sampling data, t is the stockline detection time, and l is the charge surface detection height. In the same way, the finite set of time series of mechanical probes is described as follows:(2)$$T={\{\left(t_{T0},l_{T0}\right),\left(t_{T1},l_{T1}\right),\cdots ,\left(t_{Tn},l_{Tn}\right)\}|}_{t_{Ti}<t_{Ti+1}\left(i=1,2,\cdots ,n-1\right)}$$
where n is the number of data sampled by the mechanical probe. Using the discrete time series joint partition method, the measurement data of the two probes can be partitioned with high precision, and the partition results are shown in Figure 4. The main steps of the algorithm are shown in Algorithm 1. After the algorithm processing, the results of the time series partition of the radar and mechanical probes can be expressed as $L=\{L_{i}|L_{i}=\{U_{i},D_{i}\},i=1,2,\cdots ,n\}$, where variable i represents the ith material charging cycle. $U_{i}=\left\{\left(t_{j},l_{j}\right)\in R|t_{\min i}\leq t_{j}\leq t_{\max i}\right\}$ represents the stockline sequence of the material feeding period in the ith cycle, and $D_{i}=\left\{\left(t_{j},l_{j}\right)\in R,\left(t_{Ti},l_{Ti}\right)\in T|t_{\max i}\leq t_{j}\leq t_{\min i+1}\right\}$ represents the stockline sequence of the idle period in the ith cycle.
**Algorithm 1:** Discrete Time Series Joint Partition MethodDetermine all extreme points and measurement time of radar data time series;Based on the measuring time point $t_{Ti}\left(i=0,1,2,\cdots ,n\right)$ of mechanical probe data, the relative position relationship between all extreme points of radar probe data and reference points is determined;According to the order principle of (minimum–datum–maximum–minimum), the time series of radar data are divided into N detection periods, and the material feeding period, idle period, and the starting and ending dividing points of the material charging period are recorded as $\left(t_{\max i},l_{\max i}\right)$, $\left(t_{\min i},l_{\min i}\right)$, and $\left(t_{\min i+1},l_{\min i+1}\right)$, respectively;**return** the detection periods N and the corresponding ending dividing points $\left(t_{\max i},l_{\max i}\right)$, $\left(t_{\min i},l_{\min i}\right)$, and $\left(t_{\min i+1},l_{\min i+1}\right)$.

### 2.2. Modeling of Radar Data Based on Maximum Likelihood Radial Basis Function Model (MLRBFM)

The sampling number of radar data of the stockline is set as n in each material charging cycle, and the time series of the radar probe data in the cycle is described as $R_{i}=\left\{\left(t_{i1},l_{i1}\right),\left(t_{i2},l_{i2}\right),\cdots ,\left(t_{in},l_{in}\right)\right\}$. Considering the material charging control strategy of blast furnaces, the workers try to maintain the fluctuation of the stockline close to the standard control stockline $l_{0}=1.5$ m in every material feeding cycle, in order to ensure the high efficiency and smooth operation of the blast furnace. Due to this material charging operation mode of field workers, it is easy to find obvious normal distribution features in statistics by statistically analyzing the stockline measurement data of the radar probe at the same time in multiple different cycles. To test this hypothesis, 14 h of continuous radar probe data were used for verification. As can be observed in Figure 5, the red dotted line represents the normal distribution data set, while the blue dot represents the radar data. The radar data are roughly evenly distributed on the diagonal line, which indicates that they conform to the normal distribution. A Kolmogorov–Smirnov check of the data shows that the degree of confidence P of the dataset at each time point is greater than 0.05, which shows that the hypothesis conforms to the normal distribution and cannot be rejected.

In order to deeply mine the intrinsic data distribution relationship of radar probe data, $m(m>10^{6})$ cycles are selected from the radar probe history database, and the sampling number of radar probes in each cycle equals n=36. At the same time, the maximum likelihood estimation method is used to obtain the normal distribution function of the radar data at each time in the selected m periods by sampling points at each time in the period. Setting the j(j=1,2,3,⋯,36)th sampling data collected by radar probe in the kth period of m periods as $l_{kj}$, using maximum likelihood estimation, the Gaussian normal distribution function of the jth sampling data of radar data in any period can be described as:(3)$$\left\{ \begin{array}{l} L_{j}\left(\mu_{j},\sigma_{j}^{2};x\right)=\frac{1}{\sqrt{2\pi }\sigma_{j}}e^{-\frac{{\left(x-\mu_{j}\right)}^{2}}{2\sigma_{j}{}^{2}}}\\ \mu_{j}=\frac{1}{m}\sum_{k=1}^{m}l_{kj}\\ \sigma_{j}{}^{2}=\frac{1}{m}\sum_{k=1}^{m}{\left(l_{kj}-\mu_{j}\right)}^{2} \end{array}\right.$$
where $\mu_{j}$ and $\sigma_{j}$ are the mathematical expectation and variance of Gaussian normal distribution of the jth data of radar probe in the period, respectively.

The change in the stockline in a blast furnace has high-dimensional spatial distribution features. In order to obtain high-dimensional spatial features of the blast furnace level with high precision, to fully reflect the spatial change law of the blast furnace level, a radar data MLRBFM can be established, based on the period division result of radar probe data and the maximum likelihood estimation of sampling points at each time in the period. The model structure is shown as:(4)$$l_{i}\left(t\right)=\sum_{j=1}^{n=36}a_{ij}exp\left(-\frac{{\left\|t-\mu_{j}\right\|}^{2}}{2\sigma_{j}{}^{2}}\right)+\varepsilon_{i}$$
where i=1,2,⋅⋅⋅,m represents the ith material charging period, $a_{ij}$ represents the j-dimensional spatial feature parameters of radar data extracted in ith period, and $\varepsilon_{i}$ represents the regression error of the MLRBFM. For the convenience of explanation, the extracted high-dimensional space feature vector of radar data is set as $\alpha_{i}={\left(a_{i1},\cdots ,a_{ij},\cdots ,a_{i36}\right)}^{T}$, the radial basis function vector is set as $\delta_{i}={\left(\delta_{i1},\cdots ,\delta_{ij},\cdots ,\delta_{i36}\right)}_{\delta_{ij}=e^{\left(-\frac{{\left\|t-\mu_{j}\right\|}^{2}}{2\sigma_{j}{}^{2}}\right)}}$, and the mechanical probe measurement data sequence $\left(t_{Ti},l_{Ti}\right)$ is combined into the above formula as a reference point and vectorized, so the maximum likelihood radial basis function model of radar data is shown as:(5)$$l_{i}\left(t\right)=\delta_{i}\cdot \alpha_{i}+\varepsilon_{i}+\left(l_{Ti}-l_{Ri}\right)$$
where $l_{Ri}$ is the radar data sampling point closest to the sampling time of the mechanical probe in the ith period. For the determination of high-dimensional spatial feature vector $\alpha_{i}$ of radar data, the least square method can be used to solve it, and its calculation formula is as follows:(6)$$\overset{}{\alpha_{i}}={\left({\delta_{i}^{t}}^{T}{\delta_{i}^{t}}^{}\right)}^{-1}{\delta_{i}^{t}}^{T}\overset{}{L}$$

$\delta_{i}^{t}$ represents the radial basis function matrix formed when the value of time variable t in radial basis function vector $\delta_{i}$ traverses $t=t_{j}$, that is, when the value of t traverses the time when the radar probe collects the j,j=1,2,3,⋯,36th data in the cycle. L is the column vector composed of the measured stockline values of the radar probe in the current period, which is described as $L={\left(l_{i1},l_{i2},\cdots ,l_{i36}\right)}^{T}$.

### 2.3. Significance Test of Radar Maximum Likelihood Radial Basis Function Model

In order to verify the significance of the MLRBFM of radar data and the validity of the extracted high-dimensional spatial features, it is necessary to test the statistical significance of the model. Taking the ith period of the MLRBFM as an example, if the model regression value at $t_{ij}$ is ${\hat{l}}_{ij}$, the model regression residual generated here can be described as $l_{ij}-{\hat{l}}_{ij}$, and its regression square sum $RSS_{i}$ and residual square sum $ESS_{i}$ can be expressed as follows:(7)$$\left\{ \begin{array}{l} RSS_{i}=\sum_{j=1}^{n}{\left({\hat{l}}_{ij}-{\bar{l}}_{i}\right)}^{2}\\ ESS_{i}=\sum_{j=1}^{n}{\left(l_{ij}-{\hat{l}}_{ij}\right)}^{2} \end{array}\right.,\bar{l}=\frac{1}{n}\sum_{j=1}^{n}l_{ij}$$
where n=36, and the unbiased estimation of F detection statistics, $R^{2}$ detection statistics, and standard deviation of MLRBFM regression analysis of radar data can be calculated by the following equation:(8)$$\left\{ \begin{array}{l} F_{i}=\frac{RSS_{i}}{ESS_{i}/\left(n-2\right)}\\ R_{i}{}^{2}=\frac{F_{i}}{F_{i}+\left(n-2\right)}\\ {\hat{\sigma }}_{i}=\sqrt{\frac{ESS_{i}}{\left(n-2\right)}} \end{array}\right.$$

For a given significance level a, the judgment condition of significance of the model is $F_{i}\geq F_{1-a}\left(1,n-2\right)$. In addition, the $t=t_{o}$ confidence interval 1−a of the calculated value ${\hat{l}}_{io}$ of the model at $t=t_{o}$ is $\left({\hat{l}}_{io}-\sigma_{o},{\hat{l}}_{io}+\sigma_{o}\right)$, where $\sigma_{o}$ is calculated by:(9)$$\left\{ \begin{array}{l} \sigma_{o}=t_{1-\alpha /2}(n-2){\hat{\sigma }}_{i}\sqrt{\frac{1}{n}+\frac{{\left(t_{io}-{\bar{t}}_{i}\right)}^{2}}{\overset{n}{\underset{j=1}{\sum }}{\left(t_{ij}-{\bar{t}}_{i}\right)}^{2}}}\\ {\bar{t}}_{i}=\frac{1}{n}\sum_{j=1}^{n}t_{ij} \end{array}\right.$$
where $R_{i}^{2}$ represents the hit rate of radar measurement data falling into the confidence interval of the model, so F and $R^{2}$ are the two main test statistics of the MLRBFM of radar data, and the higher their values, the better the significance of the model and the higher the coincidence between the model and the time series of the stockline detection data.

## 3. Structural Dynamic Self-Optimization RBF Neural Network

Under complex blast furnace conditions, there are many unknown influencing factors. The change in a blast furnace stockline presents randomness and uncertainty, which leads to the spatial features of blast furnace stockline data showing high-dimensional features. The dynamic self-optimizing RBF neural network (SDSO-RBFNN) algorithm is constructed in this paper to characterize the high-dimensional spatial features of the blast furnace stockline, integrate the advantages of the two probes, and achieve an accurate, real-time acquisition of blast furnace stockline information. Firstly, the input samples of the algorithm are composed of the high-dimensional spatial features, periodic features, and radar probe data processed by the smooth window of blast furnace charge level. Secondly, based on the feature QR decomposition direct clustering algorithm proposed in this paper, the input samples are clustered to determine the cluster center and width of the input samples. Then, based on RBF neural network, a structural dynamic self-optimization network structure is designed. By adding, merging, and pruning the hidden nodes of the network, the network structure changes with the features of the input samples, and the related parameters of the network structure are optimized and updated independently to realize the network structure self-optimization, and complete the learning and training process of the SDSO-RBFNN algorithm. Finally, the trained network is used to output the stockline information at the next moment, and the whole algorithm is realized.

### 3.1. Eigen QR Decomposition Direct Clustering Algorithm (EQRDD)

The input sample of the algorithm consists of three parts, which are the high-dimensional spatial features of each cycle, the periodic features, and the radar detection data processed by sliding window in this cycle, as shown in Equation (10).
(10)$$x_{i}(k)={(t_{\min i},t_{\min i+1},\alpha }_{i}{}^{T},l_{k-N+1}^{i},\cdot \cdot \cdot ,l_{k}^{i})$$
where $x_{i}\left(k\right)$ represents the input sample of the ith cycle and the kth window of blast furnace charge level. In a material charging cycle, the blast furnace stockline is artificially controlled above or below the standard stockline, which leads to the typical clustering phenomenon of radar data. In addition, the dusty and dynamic environment in the furnace greatly affects the accuracy and stability of radar probe measurement, which makes the measurement results fluctuate greatly, with low accuracy and a low signal-to-noise ratio. In order to deeply explore the inherent clustering features and correlation of input samples, an eigen QR decomposition direct clustering algorithm is proposed. In the algorithm, an n×m input sample matrix for m input sample datasets $X=\left[x_{1},\cdots ,x_{m}\right]$ can be formed, and is then clustered into k classes. The input sample matrix X can be expressed as the following:(11)$$XE=\left[X_{1},\cdots ,X_{k}\right],\;X_{i}=[x_{{}_{1}}^{\left(i\right)},\cdots ,x_{s_{i}}^{\left(i\right)}]$$
where E is the rotation matrix, $s_{i}$ is the number of samples contained in each class, and $X_{i}$ is the sample data vector contained in the ith class represented by the $n\times s_{i}$ matrix. For the partition ∏ of the matrix X described by Equation (11), its correlation square sum cost function can be defined as: (12)$$ss\left(\prod \right)=\sum_{i=1}^{k}\sum_{s=1}^{s_{i}}{\left\|x_{s}^{\left(i\right)}-{\bar{x}}_{i}\right\|}^{2},{\bar{x}}_{i}=\sum_{s=1}^{s_{i}}\frac{x_{s}^{\left(i\right)}}{s_{i}}$$
where ${\bar{x}}_{i}$ is the average vector of the ith class sample data, the vector $e=(\overset{s_{i}}{\overset{︷}{1,\cdots 1,\cdots ,1}})$ is introduced, and by means of the matrix, the Hilbert–Schmidt norm is shown in Equation (13).
(13)$${\left\|X\right\|}_{F}=\sqrt{Tr\left(X^{T}X\right)}$$

Equation (12) can be deformed into:(14)$$\begin{array}{l} ss\left(\prod \right)=\sum_{i=1}^{k}{\left\|X_{i}-{\bar{x}}_{i}e^{T}\right\|}_{F}^{2}\\ \;\;\;=\sum_{i=1}^{k}{\left\|X_{i}\left(I_{si}-ee^{T}/s_{i}\right)\right\|}_{F}^{2} \end{array}$$

Note that $I_{si}-ee^{T}/s_{i}$ is the projection matrix, so the following formula holds constant:(15)$${\left(I_{si}-ee^{T}/s_{i}\right)}^{2}=I_{si}-ee^{T}/s_{i}$$

In combination with Equations (13)–(15), Equation (12) can be further changed into the following form:(16)$$ss\left(\prod \right)=\sum_{i=1}^{k}\left(Tr\left(X_{{}_{i}}^{T}X_{i}\right)-\left(\frac{e^{T}}{\sqrt{s_{i}}}\right)X_{{}_{i}}^{T}X_{i}\left(\frac{e}{\sqrt{s_{i}}}\right)\right)$$

Setting the orthogonal matrix U of m×k as follows:(17)$$U=\begin{array}{c} s_{1}\\ s_{2}\\ \vdots \\ s_{k} \end{array}\left[\begin{array}{cccc} \frac{e}{\sqrt{s_{1}}} & & & \\ & \frac{e}{\sqrt{s_{2}}} & & \\ & & \ddots & \\ & & & \frac{e}{\sqrt{s_{k}}} \end{array}\right]$$

Equation (16) can be simplified into:(18)$$ss\left(\prod \right)=Tr\left(X^{T}X\right)-Tr\left(U^{T}X^{T}XU\right)$$

Obviously, the best strategy of dividing the input sample matrix X into k class is equivalent to the partition strategy, which minimizes the correlation square sum cost function min ss(∏) described in Equation (12). Considering that $Tr\left(X^{T}X\right)$ is determined by the sample space, the min ss(∏) problem is equivalent to the optimization problem described as:(19)$$\underset{U^{T}U=I_{k}}{max}\;Tr\left(U^{T}X^{T}XU\right)$$

According to the Ky Fan theorem, for the real symmetry matrix $A=X^{T}X$ with eigenvalues, $\lambda_{1}\geq \lambda_{2}\geq \cdots \lambda_{m}$ is its eigenvalue and $\left(\nu_{1},\nu_{2},\cdots ,v_{m}\right)$ is its eigenvector. Under the constraint of $U^{T}U=I_{k}$, the following equation holds constant:(20)$$\lambda_{1}+\lambda_{2}+\cdots \lambda_{k}=\underset{U^{T}U=I_{k}}{max\;}\;Tr\left(U^{T}AU\right)\;$$

The optimal matrix $U^{*}$ can be calculated by: (21)$$U^{*}=\left[\nu_{1},\cdots ,\nu_{k}\right]H$$
where H is any orthogonal matrix. Then,
(22)$$\begin{array}{l} min\;ss\left(\prod \right)\\ \;\geq Tr\left(X^{T}X\right)-\underset{U^{T}U=I_{k}}{max}\left(U^{T}X^{T}XU\right)\\ \;=\sum_{i=k+1}^{\min\left\{m,n\right\}}\sigma_{i}^{2}\left(X\right) \end{array}$$
where $\sigma_{i}\left(X\right)$ represents the ith eigenvalue of matrix X. 

In order to quickly determine the optimal matrix $U^{*}$ and synchronously divide the m input sample datasets into optimal k classes, it is assumed that the classification result of the input sample matrix X described by Equation (11) is the optimal classification, which can minimize ss(∏) of Equation (12). Each submatrix $X_{i}$ represents a class, then the following formula holds:(23)$$X^{T}X=\left[\begin{array}{cccc} X_{{}_{1}}^{T}X_{1} & 0 & \cdots & 0\\ 0 & X_{{}_{2}}^{T}X_{2} & \cdots & 0\\ \vdots & \vdots & \ddots & \vdots \\ 0 & 0 & \cdots & X_{{}_{k}}^{T}X_{k} \end{array}\right]$$

The largest eigenvector of $X_{i}^{T}X_{i}$ is set to be $y_{i}$ which satisfies the following equation:(24)$$A_{i}^{T}A_{i}y_{i}=\mu_{i}y_{i},\;\left\|y_{i}\right\|=1,\;i=1,\cdots ,k$$

Constructing the matrix $Y_{k}$ as follows:(25)$$Y_{k}=\begin{array}{c} s_{1}\\ s_{2}\\ \vdots \\ s_{k} \end{array}\left[\begin{array}{cccc} y_{1} & & & \\ & y_{2} & & \\ & & \ddots & \\ & & & y_{k} \end{array}\right]$$

According to Davis−Kahan sin(Θ) theorem, it can be deduced that the following equation holds:(26)$${ϒ}_{k}\equiv \left[\gamma_{1},\cdots ,\gamma_{k}\right]=Y_{k}V+ο\left(\left\|E\right\|\right)$$
where A is the eigenvector of matrix B and satisfies the requirements of the following equation:(27)$$X^{T}X\gamma_{i}=\lambda_{i}\gamma_{i},\;\lambda_{1}\geq \;\lambda_{2}\geq \;\cdots \geq \;\lambda_{m}$$

Matrix $V=\left[v_{1},\cdots ,v_{k}\right]$ is an orthogonal matrix of k×k; approximating Equation (26), ignoring ο(‖E‖) term, and taking transposition on both sides of the equation, it is reduced to: (28)$${ϒ}_{k}^{T}=V^{T}Y_{k}^{T}$$

Expanding the above equation:(29)$${ϒ}_{k}^{T}=(\underset{cluster\;1}{\underset{︸}{y_{11}v_{1},\cdots ,y_{1s_{1}}v_{1}}},\cdots ,\underset{cluster\;k}{\underset{︸}{y_{k1}v_{k},\cdots ,y_{ks_{k}}v_{k}}})$$

Obviously, Equation (28) indicates that the best classification strategy of input sample matrix X can be determined by the orthogonal linear transformation of matrix ${ϒ}_{k}^{T}$, which is composed of the first k maximum eigenvectors of matrix $X^{T}X$, and Equation (29) gives a discriminant formula for judging which class the input sample belongs to. It is easy to prove that the orthogonal transformation of matrix ${ϒ}_{k}^{T}$ into $Y_{k}^{T}$ can be realized by QR decomposition of matrix ${ϒ}_{k}^{T}$:(30)$${ϒ}_{k}^{T}P=QR=Q\left[R_{11},R_{12}\right]$$
where P is a permutation matrix, Q is an orthogonal matrix of k×k, and $R_{11}$ is an upper triangular matrix of k×k. By calculating ${ϒ}_{k}^{T}$, the category discrimination matrix $\overset{⌢}{R}$ can be calculated by the following formula:(31)$$\overset{⌢}{R}=R_{11}^{-1}\left[R_{11},R_{12}\right]P^{T}=\left[I_{k},R_{11}^{-1}R_{12}\right]P^{T}$$

According to the row index number of the element with the largest absolute value of the column corresponding to matrix $\overset{⌢}{R}$, the cluster category to which each data vector belongs can be determined. After determining the cluster category to which each input sample belongs, the sample set $I\left(C_{i}\right)$ of any category i can be determined, and the calculation formula of each cluster center vector $c_{i}$ is given.
(32)$$c_{i}={\sum_{x_{s}\in I(C_{j})}x_{s}/s_{i}}_{i=1,2,\cdots ,k}$$
where $s_{i}$ represents the number of elements in set $I\left(C_{i}\right)$.

### 3.2. Structure Dynamic Self-Optimizing RBF Neural Network (SDSO-RBFNN)

Due to the complexity of blast furnace working conditions, and the harsh environment, there is a strong non-linear correlation between radar probe data and mechanical probe data. In order to effectively fuse the data of the two probes, and improve the accuracy of stockline measurement on the premise of ensuring the real-time performance of the algorithm, the SDSO-RBFNN network, with a network structure that efficiently adapts data features and automatically optimizes network nodes and parameters, is proposed. Its network structure is shown in Figure 5, and the mathematical key model of the network is shown as:(33)$$\{{}_{\varphi_{j}\left(x_{i}\left(k\right)\right)=exp\left(-\frac{{\left\|x-c_{j}\right\|}^{2}}{\vartheta_{j}^{2}}\right)\;,\;j=1,2,\ldots ,K}^{\begin{array}{l} f\left(x_{i}\left(k\right)\right)=y_{i}\left(k\right)=\sum_{j=1}^{m}\omega_{j}\varphi_{j}\left(x_{i}\left(k\right)\right) \end{array}}$$
where $x_{i}\left(k\right)$ is the kth input sample data of the ith cycle, $y_{i}\left(k\right)$ is the kth output stockline data measured by the network in the ith cycle, $\varphi_{j}\left(x_{i}\left(k\right)\right)$ is the Gaussian excitation function of the jth hidden node, $c_{j}$ is the center of the jth basis function, $\vartheta_{j}$ is the width of the jth basis function, K is the number of hidden nodes, and $\omega_{j}$ is the output weight.

The number of hidden nodes, the center, and the width of the basis function can be determined by real-time clustering with the EQRDD clustering algorithm for the input sample set X. The number of clustering classes K is the number of hidden nodes, the cluster center vector $c_{j}$ is the center of the basis function, and the width $\vartheta_{j}$ of the basis function can be determined by: (34)$$\{{}_{\begin{array}{l} \vartheta_{j}=\frac{2d_{\max}^{j}}{3}\;(1\leq j\leq K) \end{array}}^{\begin{array}{l} G_{i}=\left\{x_{i}\left(k\right)\in I(C_{j})\right\}\\ d_{\max}^{j}=\underset{x_{i}\left(k\right)\in G_{i}}{\max}\left\{{\left\|x_{i}\left(k\right)-c_{j}\right\|}^{2}\right\} \end{array}}$$
where $d_{\max}^{j}$ is the maximum distance from cluster center to sample point. In the calculation of weights, the Gaussian basis function value matrix Λ is obtained by using Equation (35), and Λ is a m×K matrix, where m is the total number of samples, and K is the number of clusters.
(35)$$\varphi_{j}\left(x_{i}\left(k\right)\right)=exp\left(-{\frac{\left||x_{i}\left(k\right)-c_{j}|\right|}{\vartheta_{j}^{2}}}^{2}\right)$$

Then, the mechanical probe data corresponding to the input sample is formed into a matrix $YW={\left[\Lambda^{T}\Lambda \right]}^{-1}\Lambda^{T}Y$, and the weight vector W can be determined by the following formula:(36)$$W={\left[\Lambda^{T}\Lambda \right]}^{-1}\Lambda^{T}Y$$

Considering that the SDSO-RBFNN network can change with the increase in samples, automatically update, merge, and generate hidden nodes in the network, automatically adjust the number of hidden layer nodes, transform the network structure, and independently optimize the network structure parameters online, it is also necessary to clarify the strategy of the SDSO-RBFNN network to independently optimize and update the network structure. The strategy is shown in Figure 6, and its specific steps are as follows:

Step 1: Assuming that j hidden layer nodes already exist at time k, when the kth data sample of the ith period enters the network, it is necessary to compare the similarity between the kth input vector and the center vector of the existing j hidden nodes to find the hidden layer node $c_{l}$ with the largest similarity, and set the maximum similarity as:(37)$$s_{\max}\left(x_{i}\left(k\right),c_{j}\right)=s\left(x_{i}\left(k\right),c_{l}\right)$$

Step 2: Judging the size between $s\left(x_{i}\left(k\right),c_{l}\right)$ and threshold b, if $s\left(x_{i}\left(k\right),c_{l}\right)>b$, it is considered that the current network can complete the learning of new data, the number of hidden layer nodes remains unchanged, and the parameters of the lth hidden layer node are adjusted using Equation (38).
(38)$$c_{l}=\frac{c_{l}+x_{i}\left(k\right)}{2}$$

Otherwise, it is considered that the kth input vector cannot activate any existing hidden layer node, and a new hidden node requires to be added, that is,
(39)$$\left\{ \begin{array}{l} j=j+1\\ c_{j+1}=x_{i}\left(k\right) \end{array}\right.$$

Threshold b can control the classification accuracy of the network. When b is larger, the classification accuracy is higher, and when b is smaller, the classification accuracy is lower. According to different requirements, different values can be taken for b to satisfy the requirements.

Step 3: The distance between all hidden layer nodes in pairs is checked. If the distance between two hidden layer nodes is less than d, the node with the largest node width is taken as the merged new node, and the existing hidden nodes are trimmed by Equation (40).
(40)$$c_{j}=\left\{ \begin{array}{l} c_{i}\;if\;dist(c_{i},c_{j})<d\;and\;\vartheta_{i}\;>\vartheta_{j}\\ c_{j}\;otherwise \end{array}\right.$$

The sample similarity calculation equation in the above steps is shown as:(41)$$s\left(x^{p},x^{q}\right)=1-\frac{dist\left(x^{p},x^{q}\right)}{\left\|x^{p}\right\|+\left\|x^{q}\right\|}$$
where $x^{p}$ and $x^{q}$ are any two different input sample vectors, $dist(x^{p},x^{q})$ is the distance between samples, and ‖·‖ is the length of sample vector. To ensure that the distance between samples is reasonable and fully reflects the similarity of samples as much as possible, $dist(x^{p},x^{q})$ is defined as: (42)$$dist\left(x^{p},x^{q}\right)=\sqrt{\sum_{h=1}^{p}w_{h}{\left(x_{h}^{p}-x_{h}^{q}\right)}^{2}}$$
where p is the dimension of the input vector, $x_{h}^{p}$ represents the hth variable value of the pth sample, and $w_{h}$ is the weight of the hth dimension, which is determined by: (43)$$\left\{ \begin{array}{l} I\left(X,Y\right)=\int_{y}\int_{x}p\left(x,y\right)\log\frac{p\left(x,y\right)}{p\left(x\right)p\left(y\right)}dxdy\\ w_{h}=\frac{I_{h}}{\sum_{h=1}^{p}I_{h}} \end{array}\right.$$
where $I_{h}$ is the mutual information value of the hth variable, and p(x,y) is the joint probability density of random variables X and Y.

In summary, the algorithm steps of SDSO-RBFNN are as follows:

Step 1: According to the features of input samples, the values of category discrimination radius b and node distance d are set;

Step 2: When the first sliding window radar data of the first material charging cycle are sampled, the first input sample $x_{1}\left(1\right)$ is constructed according to Equation (10), a first class is formed, and the number of classes K=1, the clustering centers $C=\left\{c_{1}\right\}$ and $c_{1}=x_{1}\left(1\right)$, and the first hidden node $h_{1}$ of the hidden layer of the network are generated;

Step 3: When the data of the k input sample $x_{1}\left(k\right)$ in the ith material charging cycle is sampled, the number of samples m=k, k=2,3,⋯, there are K−1 cluster categories, and the cluster center $C=\left\{c_{1},c_{2},\cdots ,c_{K-1}\right\}$ and $h_{P},p=1,2,3,\cdots ,K-1$ hidden nodes are formed. Setting the input sample data matrix as $X=\left(x_{i}\left(1\right),\cdots ,x_{i}\left(m\right)\right)$, when $min{\left\|x_{i}\left(k\right)-c_{j}\right\|}^{2}{}_{c_{j}\in C}<b$ is established, the current clustering results are kept and network structure unchanged, Step 7 is next. Otherwise, the number of hidden layer nodes increases by 1 and Step 4 is initiated;

Step 4: Based on the input sample data matrix X, the matrix $X^{T}X$ is calculated, the first K maximum eigenvectors of $X^{T}X$ are used to form the matrix ${ϒ}_{k}^{T}$, and the matrix ${ϒ}_{k}^{T}$ is QR decomposed. Then, the class discriminant matrix $\overset{⌢}{R}$ is calculated based on Formula (31), and all samples are reclassified to obtain a new class set $I=\left\{I\left(C_{1}\right),I\left(C_{2}\right),\cdots ,I\left(C_{K}\right)\right\}$;

Step 5: The cluster center vector set $C=\left\{c_{1},c_{2},\cdots ,c_{K}\right\}$ is updated based on Equation (32), then the basis function centers and widths of K hidden nodes on the hidden layer of the network are synchronously updated based on Equation (34), while all the hidden nodes are merged and pruned, based on Equation (40);

Step 6: The network weight vector is calculated and updated using Equation (36) to complete the establishment and training of SDSO-RBFNN;

Step 7: Using Equation (33), $x_{i}\left(k\right)$ is taken as the model input, and the current stockline information $y_{i}\left(k\right)$ is calculated and obtained in real time, so as to realize high-precision, real-time measurement of the blast furnace stockline.

Although the number of neurons in SDSO-RBFNN K is always in the process of dynamic change, it eventually changes dynamically within a certain range. After long-term experiments, we find that K tends to be in the order of $10^{2}$ under long-term stable normal working conditions, but when abnormal working conditions occur, the number of K increases greatly and decreases after the abnormal working conditions end.

## 4. The Simulation and Industrial Verification Results

For the training of the network model, the hardware environment included Intel(R) Xeon(R) CPU E5-2620 v3 @ 2.40GHz, NVIDIA Geforce, and one RTX2080 graphics card. The language chosen was python3.7.4, the platform used was PyTorch 1.9.0, and the CUDA version was 11.4.0. All our simulation diagrams were drawn based on MATLAB. In order to select a suitable maximum likelihood radial basis function model, and verify the effectiveness of the proposed real-time measurement method for blast furnace stockline, the actual measurement data of a mechanical probe and a radar probe in the same area on the north side of a 2650 m^3^ blast furnace in a steel plant were used for simulation. The sampling interval of mechanical probe was 3–6 min, and the number of samples was 3393. The sampling interval of radar probe was 10 s, the number of samples was 78,816, the width of the sliding window was set to N=10, according to experience, and then the radar data were adjusted to a 3393×10 matrix.

### 4.1. Selection and Comparison of Maximum Likelihood Radial Basis Function Model

In this paper, a maximum likelihood radial basis function model (MLRBFM) was proposed to mine the intrinsic correlation of blast furnace stockline radar data and extract the high-dimensional spatial features of stockline changes. In the MLRBFM algorithm, the cycle number m has a great influence on the extraction accuracy of high-dimensional spatial features. If the cycle number m is too small, the internal data features of the radar data cannot be fully extracted, and the accuracy of the model decreases. An excessive cycle number m leads to the high complexity of the model, serious over-fitting of the model, and the accuracy also declines. In order to find the best matching cycle number m, an ablation experiment, as shown in Figure 7, was carried out. On the premise of changing only the cycle number m, the MRE and RMSE curves of the SDSO-RBFNN method based on MLRBFM with different m values were drawn under normal and abnormal working conditions. It can be seen from Figure 7 that with a gradual increase in the m value, the accuracy of extracting high-dimensional spatial features of radar data by the algorithm is improved, and the detection error of the SDSO-RBFNN method is gradually reduced. When $m=10^{6}$, both MRE and RMSE achieve the minimum value, and the algorithm proposed in this paper basically reaches the minimum value compared with the reference data. When the m value increases further, the model complexity and over-fitting phenomenon become the core factors leading to the decline in algorithm accuracy, and the accuracy of the algorithm declines rapidly; when m is greater than $10^{8}$, the complexity and over-fitting of the model are saturated, and the accuracy of the algorithm is improved slightly with the increase in data quantity. When m is greater than $10^{14}$, due to excessive accumulation of abnormal sample data, the accuracy of the algorithm decreases with the increase in abnormal samples, that is, with the continuous increase in m, the accuracy of the algorithm continues to decline. Therefore, $m=10^{6}$ is selected as the best cycle number.

Figure 8 is a comparison chart of stockline measurement trends with different cycle numbers, including normal and abnormal working conditions of the blast furnace. Figure 8 not only shows the measured values of the mechanical probe and radar probe, but also draws the curves of the stockline measured by the proposed algorithm when m is $10^{6}$, $1/2\times 10^{6}$, and $1/3\times 10^{6}$, respectively. It can be seen from Figure 8 that when the blast furnace is in normal working condition, the blast furnace stockline data usually fluctuate around the standard control stockline of 1.5 m. At this time, the closer the value of m is to $10^{6}$, the higher the accuracy of the algorithm for measuring the stockline, the smaller the fluctuation, and the better the consistency with the measured value of the mechanical probe. When the blast furnace is in an abnormal working condition shown in Figure 8, the real stockline of the blast furnace measured by the mechanical probe is greater than the standard control stockline of 1.5 m. Figure 8 shows that the stockline is close to 2 m. At the same time, due to dust and the high-speed flow and high-frequency fluctuation of airflow in the top of the furnace, the radar measurement data also follow frequent and large-scale jumps, and the measured values are often far from those measured by the mechanical probe. The average absolute error of the two measurements in Figure 8 is about 0.5 m. At this time, by observing the algorithm measurement curves based on different m, it is found that when m is too small, the accuracy of the MLRBFM method for high-dimensional spatial feature extraction of radar measurement data decreases, which means that the SDSO-RBFNN method is unable to fit the change of radar probe measurement data well, resulting in the change in stockline measurement data being limited in a narrow range. Obviously, this is inconsistent with the actual blast furnace material charging conditions, and the reliability of the stockline measurement is low. With the increase in m to $10^{6}$, the SDSO-RBFNN method not only shows a high degree of consistency with the measured value of the mechanical probe, but also shows a high degree of consistency with the trend of the radar data. Therefore, it is further verified that $m=10^{6}$ is the best cycle number for the MLRBFM method. In addition, through the analysis of radar probe data and mechanical probe data, it can be seen that the change trend and fluctuation period of the blast furnace charge level plotted in Figure 8 are highly similar to the radar data. This shows that SDSO-RBFNN has high confidence in measuring blast furnace stockline data and extracting high-dimensional spatial features.

In order to further understand the influence of the extraction accuracy of high-dimensional spatial features of MLRBFM on the improvement in the accuracy of SDSO-RBFNN, Figure 9 provides a graph of stockline measurements by the proposed algorithm when the variance and mean value of radial basis function are randomly given in MLRBFM. Comparing the measurement results of SDSO-RBFNN and random RBFNN with random parameters of the radial basis function, it can be seen that when the fluctuation of the blast furnace stockline is small and gentle, the measured values of the above two methods are basically consistent with the measured stockline of the mechanical probe. When the blast furnace stockline jumps and fluctuates greatly, because the radial basis function parameters in MLRBFM are given randomly, MLRBFM does not fully explore the inherent distribution law of radar data, so the accuracy of extracting high-dimensional spatial features of the blast furnace stockline by MLRBFM is poor. This leads to the large deviation between the measured results of random RBFNN and the true values, and it is difficult to track the changing trend of radar data and coincide with the measured values of the mechanical probe. However, SDSO-RBFNN, which extracts the high-dimensional space features of stockline accurately, exhibits a good similarity to the mechanical probe measurements. Therefore, the high-precision extraction of high-dimensional space features of the blast furnace based on MLRBFM has significance for the accuracy of the stockline measurement algorithm in this paper. 

### 4.2. Verification of the Blast Furnace Stockline Detection Based on SDSO-RBFNN

In order to fully verify the accuracy, continuity, and reliability of real-time measurements of the blast furnace stockline by the SDSO-RBFNN algorithm proposed in this paper, Figure 10 is a comparison chart of blast furnace stockline measured by a traditional RBFNN method and the SDSO-RBFNN method under normal working conditions. From the measured data of the mechanical probe, the stockline is basically around 1.5 m of a standard control stockline, and the material charging cycle length is basically fixed. Radar probe data fluctuates around the mechanical probe data, but at the measuring points of the mechanical probe, the measured values of the radar probe are slightly deviated, which shows that the accuracy of the radar probe data is not high. Analyzing the measurement results of traditional RBFNN, this method extracts the time-dimension periodic features of radar data, and keep a certain consistency with the change trend of radar data. However, the traditional RBFNN cannot track the change range of radar detection data effectively, which shows a strong inhibition to the fluctuation of stockline, so that the detected stockline cannot deviate greatly from the standard control stockline of the blast furnace. However, in the actual material charging process, it is normal for the stockline to deviate from 1.5 m greatly, which makes the error of this method large when the stockline fluctuates greatly. In addition, the blast furnace stockline measured by the traditional RBFNN method is generally higher than that measured by mechanical probe, which is caused by the fact that this method refers to radar probe data unilaterally, ignores the correlation between the radar probe and mechanical probe data, and fails to effectively use accurate mechanical probe data to correct the deviation of the radar probe data. In the SDSO-RBFNN method, the time features and accurate high-dimensional spatial features of stockline data are extracted by the discrete time series joint partition method and MLRBFM method, respectively. Through the SDSO-RBFNN network, the correlation between the two kinds of detection data is analyzed, quantified, and modeled. At the same time, it has the advantages of both the radar detection and mechanical detection methods, which cannot only track the change trend of radar probe data, but also achieve a high degree of fitting to the mechanical probe data, thus, obtaining real-time and accurate stockline data, which is significantly improved compared with the traditional RBFNN in a material level measurement trend comparison.

Figure 11 is a comparison chart of the stockline trend of the blast furnace under abnormal working conditions. Under abnormal working conditions, the measured data of the mechanical probe fluctuate continuously and greatly, and deviate greatly from the standard control stockline of the blast furnace, reaching more than 2.2 m in extreme cases. At this time, due to the abnormal working conditions, the dust density in blast furnace increases sharply, the gas–powder mixed flow moves randomly and violently, and fluctuates frequently. The radar measurement data deviate significantly from the actual stockline, and the error is very large, so the radar measurement value almost loses the actual reference value. However, the traditional RBFNN method is highly dependent on the accuracy of the radar probe data, which makes it impossible to fit the measured data of the mechanical probe under abnormal working conditions. Moreover, the method itself restrains the fluctuation of the stockline, which leads to the loss in tracking ability for the trend of the radar probe data with a large amplitude and multiple movements, and, thus, its reliability and accuracy are low. The proposed SDSO-RBFNN method performs well under abnormal working conditions. It not only shows the high consistency of the change trend of the radar detection data, but also fits well with the mechanical detection data in most cases. The fitting effect is not good only when some material surfaces show extreme changes. Therefore, the proposed SDSO-RBFNN method still has high practical value and good universality under abnormal working conditions.

Figure 12 and Figure 13 show the absolute error and relative error distribution of the stockline measured by radar, the RBFNN method, and the SDSO-RBFNN method, relative to the measured value of the mechanical probe for 3 days of continuous operation of a blast furnace. Table 1 shows the specific error distribution of the three methods. Most of the time, the blast furnace is in normal condition, but there are numerous abnormal conditions during the operation of the blast furnace. During this period, 799 accurate stockline values were measured by mechanical probe. It can be seen from the Figure 12 and Figure 13 that the relative error and absolute error of radar measurement data are large, the absolute error fluctuates within ±0.5 m, the relative error concentrates within ±20%, that is, within the red line, and the accuracy is not high. The RBFNN method is more accurate than the radar measurement method. However, because this method does not fully explore the internal correlation of radar data and extract the key high-dimensional spatial features of radar data, the measurement results are randomly distributed on both sides of the stockline data of the mechanical probe, and the absolute error fluctuates within ±0.3 m, while the relative error is evenly distributed within −18%~10%. However, in the actual blast furnace material charging process, the error of this method is still large, which is unacceptable in practical application. It can only be used as an auxiliary measurement method, and its industrial application value is low. The SDSO−RBFNN method based on MLRBFM proposed in this paper accurately extracts the high-dimensional spatial features of radar data, and at the same time, establishes an accurate model that reflects the clustering features of radar data. Figure 13 shows that the measured values of this method are widely distributed on both sides of the baseline, which indicates that it effectively learns the features of low stockline data with relatively few training samples, thus, realizing the fitting of low stockline. It is seen from Table 1 that the relative error of the measurement results of the proposed method is basically within ±5%. As the measuring method of the mechanical probe itself has a relative error of ±2%, therefore, the stockline detection method based on the proposed high-dimensional spatial features shows an accuracy similar to the mechanical probe measurement method on the premise of real-time and continuity, which meets the requirements of the blast furnace smelting material charging process, and has extremely high practical value.

Figure 14 is a 45° line diagram of the measurement results of the above three methods, where the red line is the accurate value, which shows the accuracy of the three stockline measurement methods in a more intuitive way. As seen from Figure 14, the data distribution of the radar probe is clustered, concentrated and distributed above the measured stockline, and the deviation is large. The deviation comes from the interference of the high concentration of dust and airflow, and the interference increases with the stockline deviating from the standard control stockline of the blast furnace. That is, when the stockline of the blast furnace is higher than 2.0 m, the radar data deviates greatly from the standard line and jumps frequently, resulting in extremely low accuracy. When the RBFNN method is near 1.5 m of the standard stockline, that is, at the midpoint of the standard line, the measured values can be well gathered on both sides of the standard line, which indicates that the method has good measurement accuracy when the blast furnace condition is stable. However, when the stockline moves to both ends of the standard line, that is, when the working condition of blast furnace deteriorates gradually, this method cannot fully reflect the intrinsic clustering features of the radar data. This is because it does not extract the spatial features of radar data, resulting in a sharp decline in its measurement accuracy. Therefore, the image presents a wedge shape with divergence at both ends and aggregation in the middle; therefore, the universality of the method is poor, and it is difficult to use it in the actual process for an extended period of time. The proposed SDSO-RBFNN method has little difference with the measured stockline data of the mechanical probe under normal and abnormal working conditions, and the measured values are closely distributed on both sides of the reference line, which is not only accurate, but also suitable for various abnormal working conditions of the blast furnace.

Selecting the 12 h real operation data of the proposed method, the VS-RBFNN [20] method, and the ESST-RBFNN [21] method on a 2650 m^3^ blast furnace, a long-term operation effect diagram, as shown in Figure 15, is drawn to compare the detection accuracy of the three detection methods. Obviously, the VS-RBFNN method is similar to the traditional RBFNN method, showing inhibition to the predicted stockline, and the predicted stockline fluctuates slightly at 1.55 m. Compared with the real stockline, the prediction of the stockline under normal working conditions is on the high side, and the prediction of stockline under abnormal working conditions has great deviation. It can be seen that, although this method fits the changing trend of radar gauge data well, the deviation correction effect of the mechanical probe data on the predicted value and the real value is poor. The prediction results of the ESST-RBFNN method at normal and high stockline are basically consistent with the data of the mechanical probe. However, when predicting the low stockline, the predicted stockline frequently changes violently, which is caused by the failure of this method to extract the spatial features of radar data effectively, and the poor stability of the clustering algorithm. Under normal working conditions, the predicted stockline of the proposed method is basically equal to the actual material level, and under abnormal working conditions, the deviation between the predicted stockline and the actual stockline is small, and the predicted stockline data change smoothly. This shows that this method has higher accuracy and stronger stability than the other methods.

In order to further verify the statistical correctness and effectiveness of the stockline measurement method in this paper, taking the mechanical probe data as the standard value, the stockline data, including normal and abnormal working conditions and lasting for one week, are selected for testing, and the prediction results of the above method are compared with the proposed method. Figure 16 and Figure 17 are a box-whisker plot and a histogram of the relative errors of the prediction results of the three methods, respectively. As can be seen from Figure 16, the relative error outliers of the proposed method, that is, the red dots in the figure, are less than the relative error outliers of other methods. As can be seen from Figure 17, the relative errors of the proposed method are more concentrated, and most of them are within ±5%. This shows that, compared with other existing methods, this method is more stable and has higher overall accuracy. In addition, these data are used to calculate the evaluation indexes of the three methods. As shown in Table 2, according to the statistical data, compared with other methods, the real-time measurement method of the blast furnace stockline based on high-dimensional spatial characteristics proposed in this paper shows significant improvement in all indexes, which provides accurate and continuous real-time blast furnace stockline information for distributors, and has very strong practical value under both normal and abnormal working conditions.

## 5. Discussion and Conclusions

### 5.1. Discussion

A high-dimensional spatial characteristic blast furnace stockline detection method based on MLRBFM and SDSO-RBFNN is proposed. Based on the blast furnace material charging process, the discrete time series joint partition method extracts the accurate time characteristics of the blast furnace material charging cycle by analyzing the stockline information. Secondly, based on the Gaussian distribution model of the radar probe data, the MLRBFM method is proposed to obtain the high-dimensional spatial features of stockline data. Finally, combined with the EQRDD algorithm, SDSO-RBFNN realizes the periodic fusion of the stockline detection data of the two probes to obtain accurate and continuous real-time stockline data. Industrial experiments and simulation results indicate that the proposed method has high accuracy, good real-time performance, strong anti-interference, and stable generalization. Under normal and abnormal working conditions, the detection results satisfy the requirements of on-site industries, and exhibit industrial application value.

### 5.2. Conclusions

The proposed method can track the blast furnace stockline information well under normal and most abnormal conditions, thus, providing continuous, high-precision, and high-stability level information for actual industrial production in real time. However, under some extreme working conditions, the data measured by radar probe are affected by the extreme environment in the blast furnace, resulting in high frequency and a high amplitude jump, which significantly reduces the accuracy of the proposed method. The main reason for this situation is that the radar data under extreme working conditions are greatly affected by the environment, lose the clustering characteristics, and do not conform to Gaussian distribution, which means the proposed method is unable to effectively extract the high-dimensional spatial characteristics of radar data, and leads to the inability to track the real material level, which needs to be further studied and improved in the future.

## Figures and Tables

**Figure 1 sensors-22-06245-f001:**
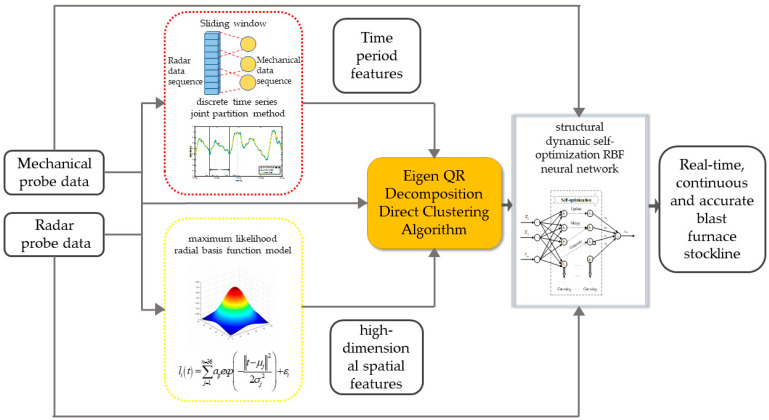
Research framework of the blast furnace stockline detection.

**Figure 2 sensors-22-06245-f002:**
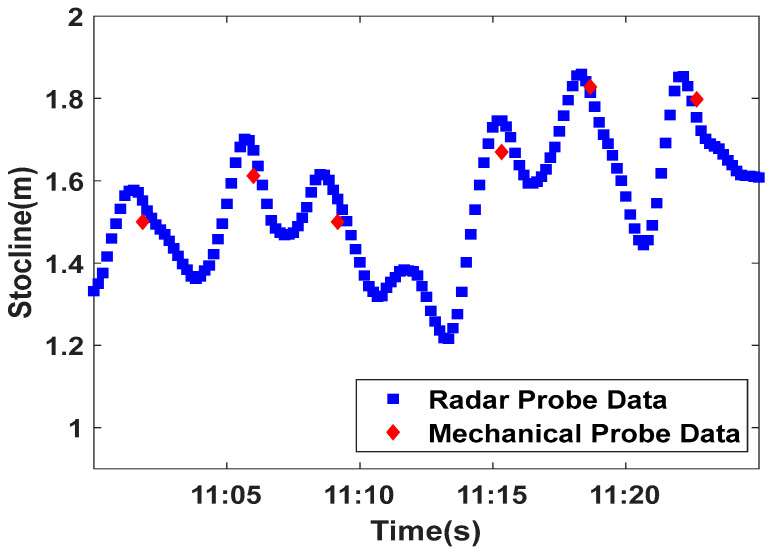
Stockline information detected by mechanical probe and radar probe.

**Figure 3 sensors-22-06245-f003:**
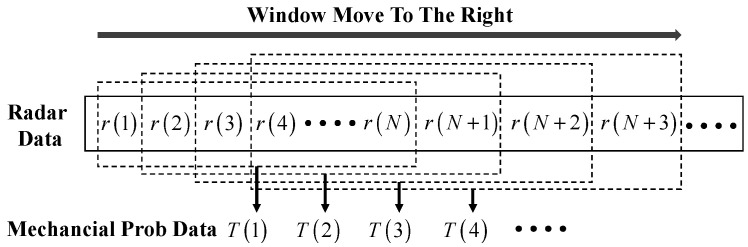
Schematic diagram of sliding window data processing.

**Figure 4 sensors-22-06245-f004:**
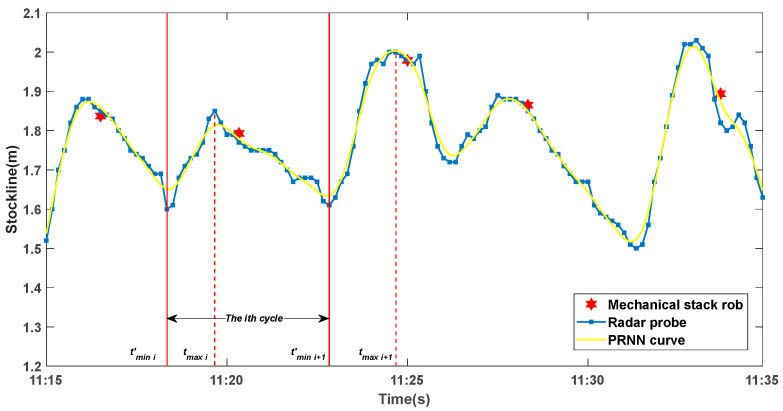
Joint partition results of discrete time series of mechanical and radar probe.

**Figure 5 sensors-22-06245-f005:**
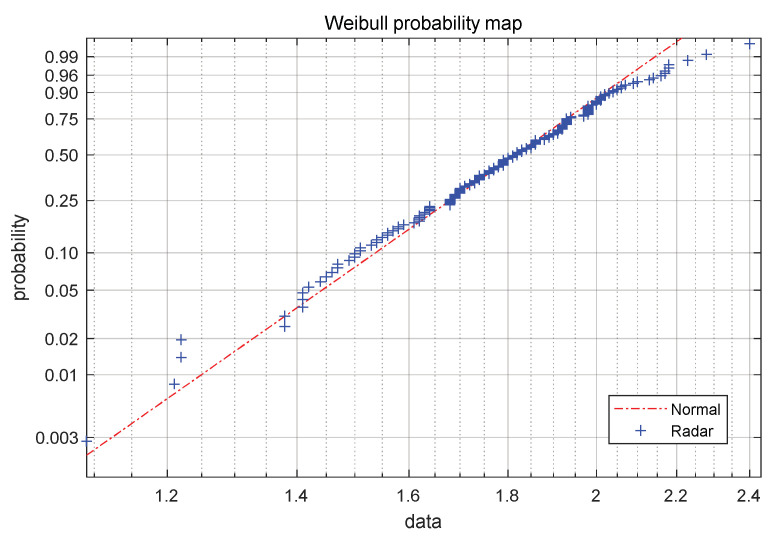
Weibull probability plot of radar probe data.

**Figure 6 sensors-22-06245-f006:**
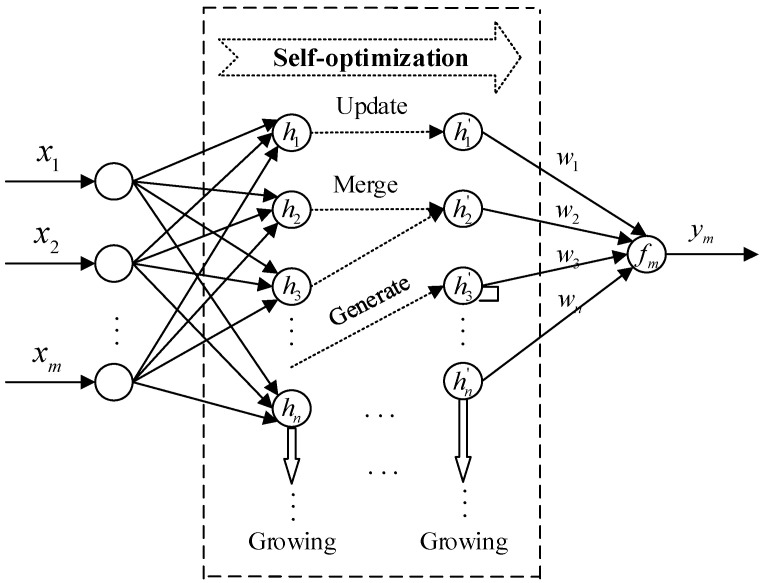
Network structure of SDSO-RBFNN.

**Figure 7 sensors-22-06245-f007:**
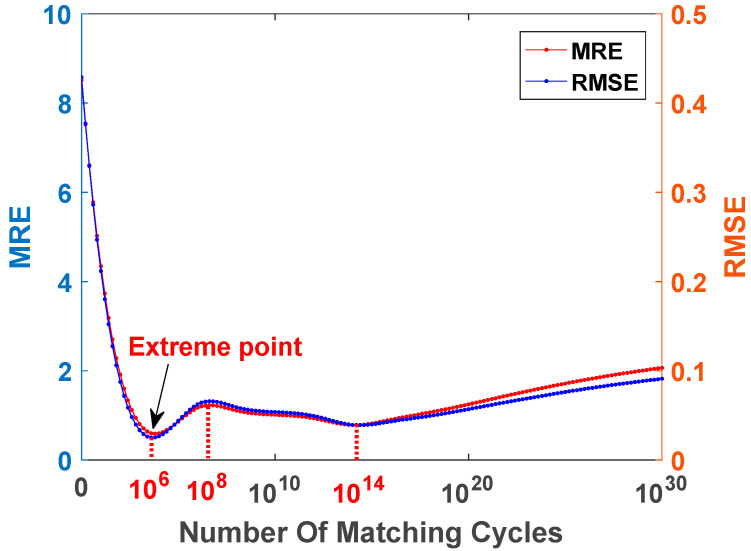
Comparison in ablation experiment with different m values.

**Figure 8 sensors-22-06245-f008:**
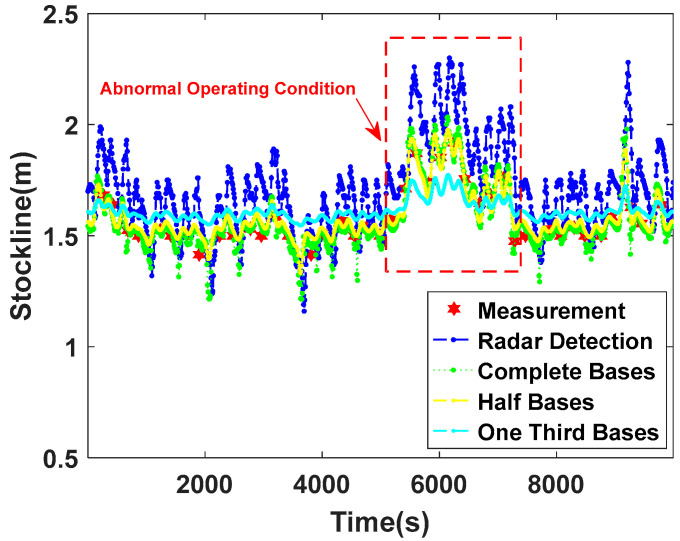
Comparison chart of stockline curve in periodic ablation experiment.

**Figure 9 sensors-22-06245-f009:**
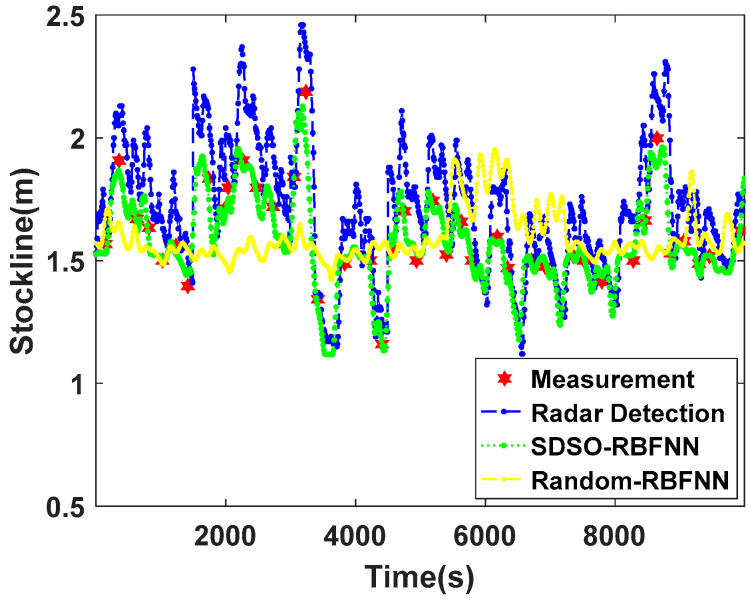
Comparisons of the selection effect of different radial basis functions.

**Figure 10 sensors-22-06245-f010:**
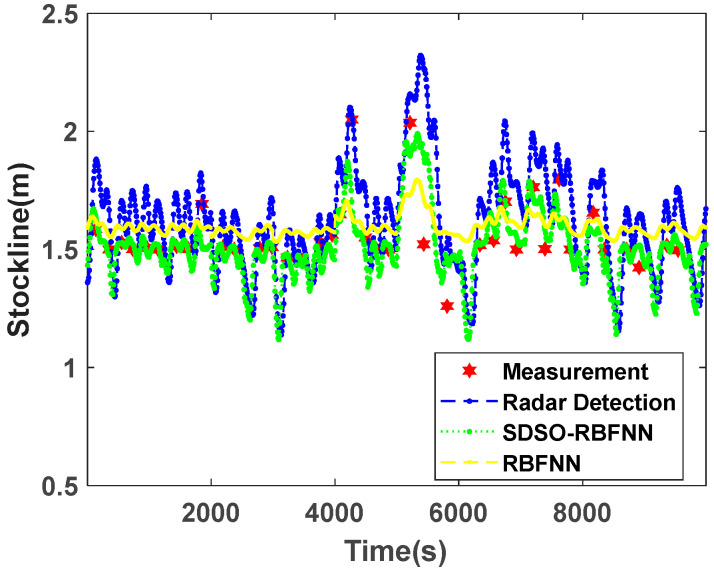
Comparison of stockline detection under normal working conditions.

**Figure 11 sensors-22-06245-f011:**
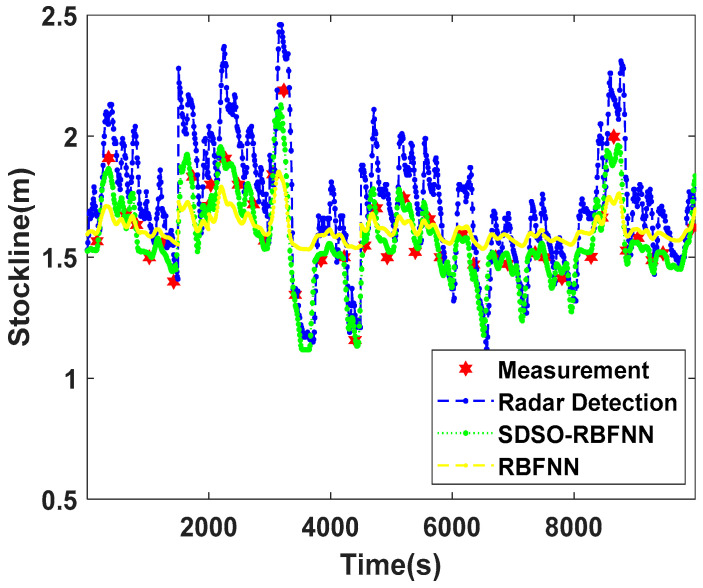
Comparison of stockline detection under abnormal working conditions.

**Figure 12 sensors-22-06245-f012:**
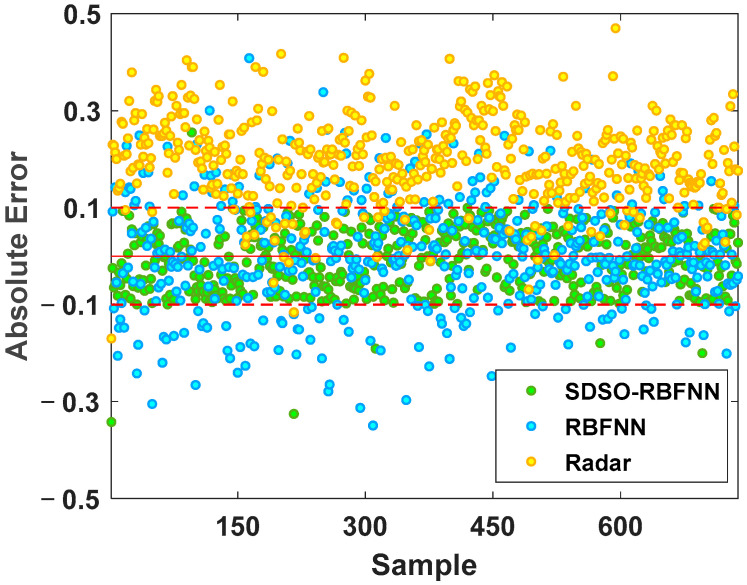
Absolute error comparison chart.

**Figure 13 sensors-22-06245-f013:**
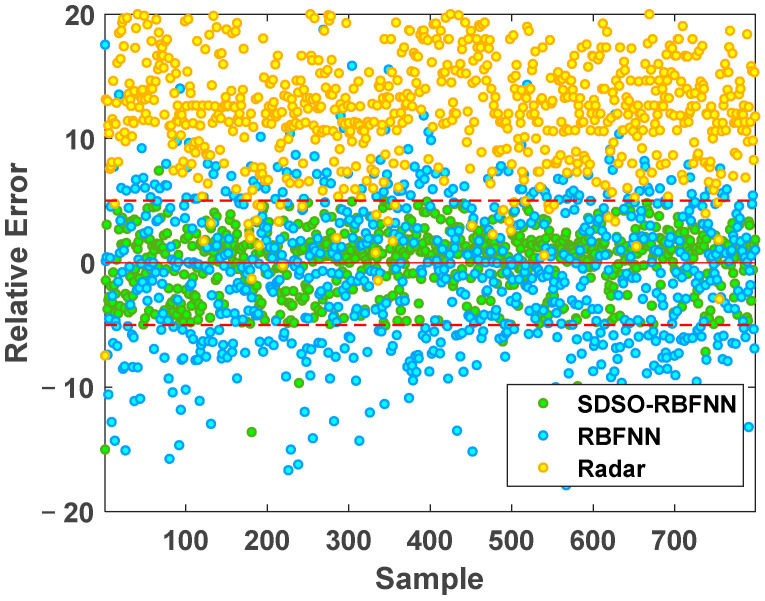
Relative error comparison chart.

**Figure 14 sensors-22-06245-f014:**
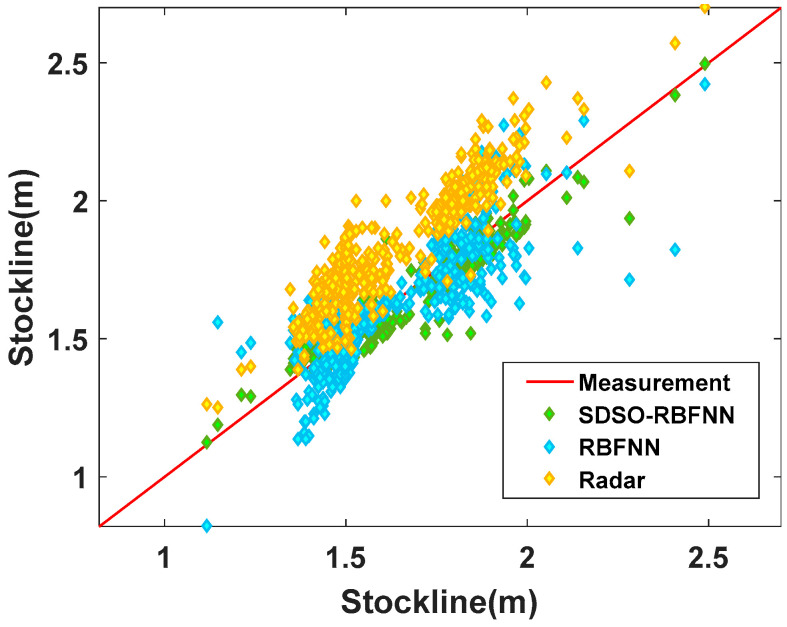
45° line diagram of the measurement effect of the methods.

**Figure 15 sensors-22-06245-f015:**
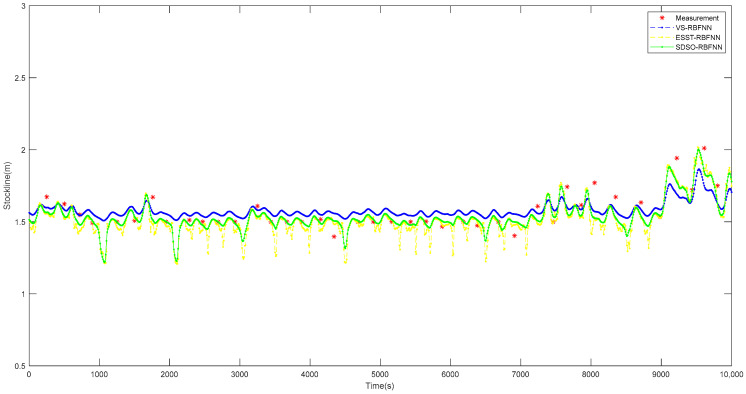
Comparison diagram of operation effect of various the detection methods.

**Figure 16 sensors-22-06245-f016:**
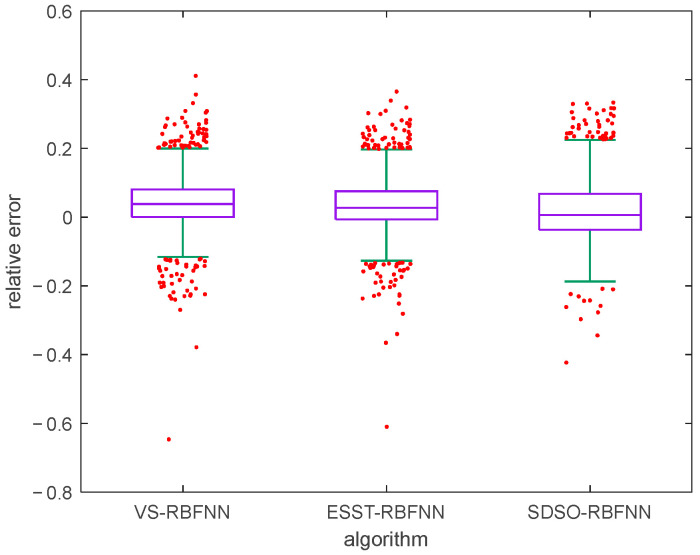
Relative error box-whisker plot of various detection methods.

**Figure 17 sensors-22-06245-f017:**
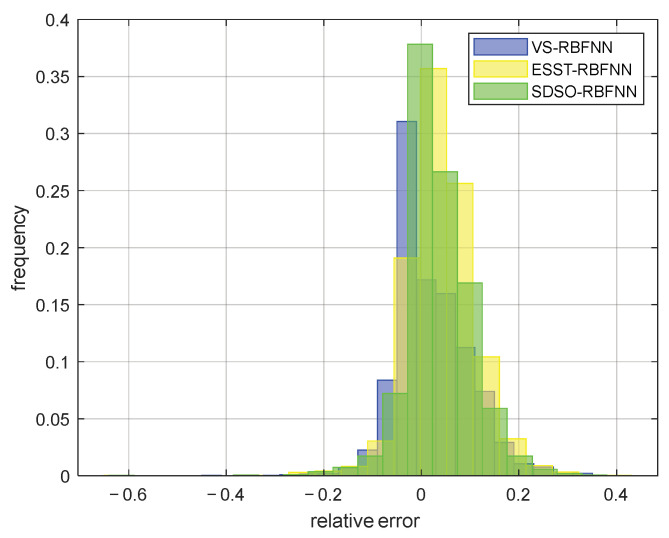
Relative error histogram of various detection methods.

**Table 1 sensors-22-06245-t001:** Statistical error comparison.

Method	Statistical Indices			
	MRE	RMSE	Error-2%	Error-5%
Radar	18.73%	0.2156	55.13%	55.13%
RBFNN	8.94%	0.1162	58.97%	67.95%
Proposed	2.73%	0.0388	92.31%	99.13%

**Table 2 sensors-22-06245-t002:** Statistical indices comparison.

Method	Statistical Indices				
	MSE	RMSE	MAE	EAP	R2
VS-RBFNN	0.0226	0.1503	0.1056	6.0227%	0.4295
ESST-RBFNN	0.0222	0.1491	0.1046	5.9823%	0.3664
Proposed	0.0200	0.1415	0.0962	5.4677%	0.3563

## Data Availability

Data available on request due to privacy. The data presented in this study are available on request from the corresponding author. The data are not publicly available for the reason that it is real-time industrial operation data. In order to be responsible for the industry, it is difficult for us to disclose these data, but you can obtain them from the corresponding author by email.

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
