# Peer review of "High-Precision Real-Time Detection of Blast Furnace Stockline Based on High-Dimensional Spatial Characteristics"

_sensors, 2022, doi:10.3390/s22166245_

Round 1

Reviewer 1 Report

General assessment:

The problem stated in the paper and its solution is fully justified by industrial needs of measurements and prediction improvement. Chapter 1 contains description proving that fact and presenting review of various alternative solutions with their advantages and drawbacks. The literature review is properly presented. The scientific tools for data analysis and extraction as well as further processing are well chosen. However, it would be good to give some implementation details, especially in case of neural networks used trained in this work e.g. in Chapter 3 Figure 5 the structure of the network is given, but details about number of layers, neurons and their common connection are not presented. Chapter 4 is very interesting, especially the deep discussion of obtained results. Therefore, I’m not sure that Chapter 5 has a proper title – it should be rather Conclusions instead of Discussion and it should be a little bit longer to point out real conclusions, not only reassumption of what was done in the work.

Minor improvements:

There are a lot of mistakes in the paper, which should be improved, mainly formatting issues like: camel case before abbreviations, caption of figures should be on the same page as the figure, dots are sometimes separated, numbers have sometimes larger fonts and sometimes the same as the text in paragraph. All mistakes which I found are marked in the paper in yellow.

Reviewer 2 Report

The paper contains an interesting application of known methods. Definitely, the theoretical part was better described than the practical part, which needs significant expansion and improvement.

My main comments:

1. No statistical tests were performed to confirm the hypotheses posed.

2. the novelty introduced should be highlighted. Have any new methods been proposed? The issues discussed have been proposed before. Have any modifications been made? Is it just the application of known methods to the detection of blast furnace stockline.

3 The comparison presented is not sufficient. The results should be compared with other state-of-the-art methods.

Other comments:

There should be a dot at the end of the sentence. This should be corrected in many places, e.g., "Firstly, the discrete time series joint partition method is used to divide the joint period of radar and mechanical probe data;"

"To ensure that the distance between samples is reasonable and fully reflects the similarity of samples as much as possible, D is defined as" and below Formula 42 does not include D.

"The weight of the h th dimension" is once denoted by w and another time by omega.

Not the best text editing. Multiple times we have unnecessary indentation before the word "where"

In addition, at the end of pages 14 and 15, a line is divided unnecessarily, also further in the text such situations occur.

The formatting of the text on page 15 needs improvement. There are no dots at the end of sentences. In one case, a line begins with a semicolon.

Overall, the experimental section is the worst written and needs improvement, writing in a more formal style. Figures should be referred by using a number. It should not be written as "condition shown in the figure".

The caption under Figures 6, 10 goes to the next page. Do not give space before a comma.

Round 2

Reviewer 2 Report

The article has been significantly revised. Still, the Authors need to review the article for editorial errors (as they occur in the article) for example, on page 3 "For example, in [20],Z. Chen et al."